# Quit attempts among tobacco users identified in the Tamil Nadu Tobacco Survey of 2015/2016: a 3 year follow-up mixed methods study

Surendran Veeraiah [ORCID],[1] Vidhubala Elangovan,[2] Jaya Prasad Tripathy,[3] Arvind Krishnamurthy,[4] Tanu Anand,[5] Mahendra M Reddy,[6] Revathy Sudhakar,[1] Niraimathi K,[2] Divyarajprabhakar Subramani,[1] Swaminathan Rajaraman,[7] Hemanth Raj Elluswami,[4] Abhay Nirgude[8]

For numbered affiliations see end of article.

**Correspondence to**
Dr Surendran Veeraiah;
v.surendran@cancerinstitutewia.org

## ABSTRACT

**Objectives** To determine current tobacco use in 2018/2019, quit attempts made and to explore the enablers and barriers in quitting tobacco among tobacco users identified in the Tamil Nadu Tobacco Survey (TNTS) in 2015/2016.

**Setting** TNTS was conducted in 2015/2016 throughout the state of Tamil Nadu (TN) in India covering 111 363 individuals. Tobacco prevalence was found to be 5.2% (n=5208).

**Participants** All tobacco users in 11 districts of TN identified by TNTS (n=2909) were tracked after 3 years by telephone. In-depth interviews (n=26) were conducted in a subsample to understand the enablers and barriers in quitting.

**Primary and secondary outcomes** Current tobacco use status, any quit attempt and successful quit rate were the primary outcomes, while barriers and enablers in quitting were considered as secondary outcomes.

**Results** Among the 2909 tobacco users identified in TNTS 2015/2016, only 724 (24.9%) could be contacted by telephone, of which 555 (76.7%) consented. Of those who consented, 210 (37.8%) were currently not using tobacco (ie, successfully quit) and 337 (60.7%) continued to use any form of tobacco. Of current tobacco users, 115 (34.1%) have never made any attempt to quit and 193 (57.3.8%) have made an attempt to quit. Those using smoking form of tobacco products (adjusted relative risk (aRR)=1.2, 95% CI: 1.1 to 1.4) and exposure to smoke at home (aRR=1.2, 95% CI: 1.1 to 1.3) were found to be positively associated with continued tobacco use (failed or no quit attempt). Support from family and perceived health benefits are key enablers, while peer influence, high dependence and lack of professional help are some of the barriers to quitting.

**Conclusion** Two-thirds of the tobacco users continue to use tobacco in the last 3 years. While tobacco users are well aware of the ill-effects of tobacco, various intrinsic and extrinsic factors play a major role as a facilitator and lack of the same act as a barrier to quit.

## INTRODUCTION

The tobacco epidemic continues to be a major public health concern with nearly

### Strengths and limitations of this study

► This is the first such study that we are aware of, to attempt a follow-up of tobacco users identified in previous survey to understand their current tobacco use status and quit attempts.

► The study involved telephone survey to contact the tobacco users.

► The mixed-methods design enabled estimation of quit rates and understanding the enablers and barriers in quitting tobacco.

► A major limitation of this study was the poor response rate of the telephonic survey which might have introduced responder bias.

► There was no objective means of verifying the responses received by telephone.

1.4 billion tobacco users worldwide. It is one of the most important preventable causes of premature death in the world claiming more than 8 million lives each year.[1 2]

To address the growing tobacco menace, the WHO Framework Convention on Tobacco Control (WHO FCTC) came into force in 2005. This international treaty has been ratified by 181 countries, and provides a roadmap for the countries to adopt and implement tobacco control measures. Article 14 of WHO FCTC mentions the dissemination of comprehensive guidelines based on scientific evidence to promote tobacco cessation. To assist in country-level implementation of the WHO FCTC, WHO also introduced a package of six technical measures termed as the MPOWER strategy, where 'O' stands for 'offer help to quit tobacco use' which is one of the key components of this strategy.

It is beyond any doubt that quitting tobacco is one of the most effective ways of saving lives and improving overall well-being. Majority

of the smokers regret ever starting to smoke and want to quit.[3] However, quitting smoking remains difficult primarily because of the addictiveness of nicotine in tobacco, along with other social and contextual factors.[4–6] It is reported that only about 3% to 5% of unassisted quit attempts are successful.[7 8]

In India, the prevalence of tobacco use in any form is 29% of all adults (42% of men and 14% of women).[9] Tobacco use contributes to nearly 10% of all deaths in the country with more than 1 million deaths in 2016.[10] According to the Global Adult Tobacco Survey 2016/2017 (GATS) in India, more than half of the current tobacco users were planning or thinking of quitting tobacco use.[9] However, we do not know how many of them actually made a quit attempt or went on to become a successful quitter. Several other large nationally representative cross-sectional studies such as GATS, National Health Family Surveys, and so on, have examined tobacco prevalence. However, these surveys are cross-sectional in nature with limited cohort-wise assessment of tobacco users and their quitting behaviour over a period of time.

A cross-sectional household tobacco survey, Tamil Nadu Tobacco Survey (TNTS), was conducted in 2015/2016 in the state of Tamil Nadu, South India, by the Cancer Institute (Women's India Association), Chennai, India, to provide reliable state and district-wise estimates of tobacco use.[11] The survey covered nearly 100 000 adults (>15 years) in all 32 districts across the state. The results of the survey showed that 5.2% were current tobacco users and about one in every five tobacco users reported to have intention to quit tobacco use in the next 1 month. But how many of them actually quit and how many of those who made a quit attempt were successful, is unknown.

In order to answer these questions, we did a follow-up of those who were identified as tobacco users in the TNTS 3 years post-survey by telephone to understand their current tobacco use status and any quit attempts made in the last 3 years. After quantitatively assessing tobacco use status, quit rates and quit attempts among previous tobacco users, it is also useful to understand the enablers that motivated and barriers they faced in quitting or attempting to quit tobacco through a qualitative approach. This will help design a tailored package of cessation and counselling intervention. Hence, a sequential explanatory mixed-method design was adopted for this study wherein the sample for qualitative study was a subset of the quantitative sample.

The specific objectives of the study were:

1. Among the tobacco users previously identified in the TNTS in 2015/2016, determine the number and proportion who could be contacted through a telephone survey in 2018/2019 and compare their characteristics with those who could not be contacted.
2. Among those contacted by telephone in 2018/2019, determine the number and proportion who (i) continue to use tobacco (smoking and/or smokeless) that is, failed or no quit attempt, (ii) made a successful quit attempt and (iii) made any quit attempt.

3. Explore the barriers and enablers in making and sustaining a quit attempt.

## METHODS
### Study design
This study employed a sequential explanatory QUAN-QUAL mixed-methods design with a cohort study design as the quantitative component followed by a descriptive qualitative component.[12] The quantitative cohort study was a follow-up of assessment of tobacco users identified during the TNTS in 2015/2016 to assess their current tobacco use and quit attempts. Following the quantitative telephone survey, the participants were categorised into three groups based on the quit attempt made and the success of the attempt. The qualitative sample was chosen from these groups proportionate to the size of the groups. Therefore, a sequential design was opted in which the qualitative component followed the quantitative one.

### Setting
#### General setting
In order to tackle the burden of tobacco use in the country, the Ministry of Health and Family Welfare launched a network of 19 tobacco cessation clinics in India in 2002 with the support from WHO. These clinics offer a wide variety of behavioural (brief advice, 5A's and 5R's, individual/group counselling) and pharmacological interventions (nicotine replacement therapy: nicotine patch, gum, inhaler, spray and non-nicotine replacement therapy: bupropion, varenicline) for tobacco cessation free of cost. A combination of behavioural support and pharmacotherapy is generally considered the best approach for treating tobacco dependence. Subsequently, the National Tobacco Control Programme was launched in 2007/2008 to be implemented by Tobacco Control Cells at the national, state and district level. Under this programme, there is also a provision of setting up Tobacco Cessation Services at the district level. India has also launched quitline (toll-free helpline service) and cessation programme wherein tobacco users can register to receive tailored cessation advice via mobile messages.

Tamil Nadu (TN) is the sixth largest state by population with about 72 million people.[13] It has 32 administrative districts. With nearly half of the population residing in urban areas, it has a high literacy rate of 80%.[14] In TN, according to GATS 2, nearly 20% use tobacco in any form, of whom 9.5% are smokers, 9.5% are smokeless tobacco users and remaining 1% use both.[9]

#### Specific setting
##### Tamil Nadu Tobacco Survey (TNTS) 2015/2016
The TNTS identified 111 363 eligible individuals aged 15 years and above, from 32 945 households across all 32 districts in TN. Of these, 99 825 individuals contacted door-to-door responded with the response rate being 89.2%. All these individuals were assessed for tobacco use,

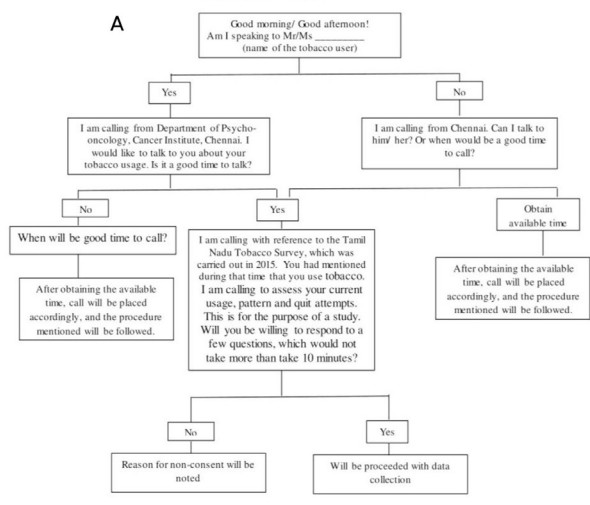

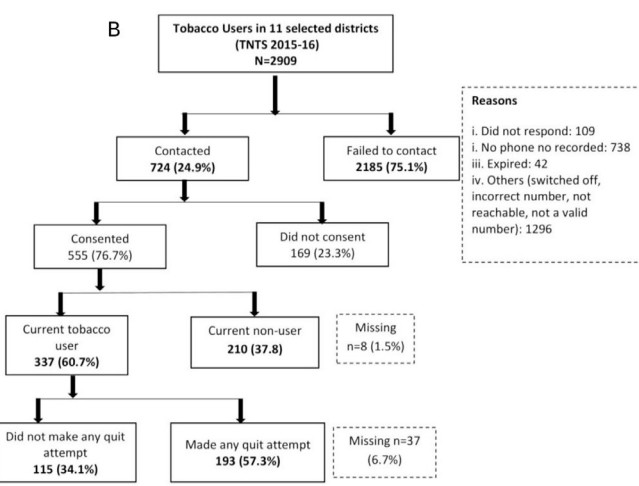

**Figure 1** (A) Standard operating procedure for making telephone calls. The telephonic calls were placed by a trained project staff. (B) Flow diagram depicting the status of current tobacco use and the pattern of quit attempts among tobacco users in six selected districts of Tamil Nadu previously identified in the Tamil Nadu Tobacco Survey (TNTS) (2015/2016).

exposure to secondhand smoke, quit behaviour, impact of pictorial warnings and other tobacco control legislations.

### Survey sampling methodology

Under TNTS, each of the 32 districts was divided into urban and rural areas, whereas Chennai city was divided into 15 zones, each zone further subdivided into slum and non-slum. The estimated sample was divided among all urban and rural areas of districts, slums and non-slum areas of zones in Chennai city using probability proportional to size sampling.[6] Data were collected during 2015/2016. The details of the survey methodology are given elsewhere.[11 15 16]

## Study population/sampling frame

The study population for both the quantitative and qualitative component included all the identified tobacco users (n=5208) from the TNTS 2015/2016. The quantitative sample was recruited by a telephone survey. The qualitative sample (n=26) is a subset of the quantitative sample.

## Sample size

Assuming that about 6% of tobacco users make a successful quit attempt, with 2% absolute precision and 80% power, sample size was calculated to be 610. Assuming 33% response rate from our previous experience (this being a telephonic survey), the final sample size was estimated to be 2025. These participants were recruited from the original TNTS survey conducted in 2015/2016 by telephonic survey.

## Data variables, sources of data and data collection
### Quantitative

Data were collected from two sources: (a) TNTS database (already collected in 2015/2016) and (b) telephonic survey (conducted in 2018/2019). A structured questionnaire was used to collect information by telephone survey with the respondents of the original TNTS survey, with the items broadly covering areas such as current tobacco usage (both smoking and/or smokeless), quit attempt(s) and their duration and their intention to quit. In addition, socio-demographic and tobacco use related variables were extracted from the TNTS. Reported tobacco users of TNTS (n=2909) in 11 districts of TN were contacted through telephone by a team of trained project staff at the Cancer Institute, Chennai. A standard operating procedure (SOP) was prepared and followed for telephone survey (figure 1). Each study participant was contacted a maximum of three times at an interval of 30 min in a day. After two calls, a standardised text message was sent stating the details of the caller and the purpose of the call. Subsequently, the tobacco user was called 30 min after the text message. This process was repeated again after 7 days (if no contact was made in the previous attempt) before labelling it as an unsuccessful contact. Response to each call was recorded by the project staff using a separate sheet as: no response, disconnected the call, number not reachable, number invalid, refused to share information, busy schedule, responded to the call and so on. Respondents who were contacted and verbally consented to participate were briefed about the purpose of the call. The questions were administered over telephone and the responses were recorded on a structured questionnaire. The telephone calls were not recorded as it might affect the responses of the participant. However, the telephone survey was monitored by an individual not associated with the current research for interviewer compliance with the protocol described above. Verbal feedback was given continuously to improve and fine-tune the process.

## Qualitative

The principal investigator (PI) (PhD in Psychology) and the co-PI (MPhil) who are trained in qualitative research methods conducted in-depth interviews (IDIs) after obtaining consent of the participants. The IDIs were conducted in regional language (*Tamil*) by telephone using an interview guide with open-ended questions related to the quit attempts made, method of quitting and motivation to quit and barriers/motivators for failed or successful quit attempts. This data was collected separately and not as a part of the quantitative data collection with all the participants, due to time constraint. The interviews were audio-recorded (after obtaining consent) and verbatim notes were also taken during the interview. Each interview lasted for around 30 min. After the interview was over, the summary of the interviews was read back to the participants to ensure participant validation. Since it was a telephone interview, no incentives were provided for the participants. A total of 8 to 10 IDIs were planned to be conducted in each district to cover those who made a successful quit attempt, failed attempt and did not made a quit attempt. It was planned to cover both smokers and smokeless tobacco users in the sample.

### Operational definitions
#### Quit attempt
Any attempt at tobacco cessation that lasts for 1 day or more than 1 day, including both self-attempt as well as attempt with professional help.[11]

#### Current tobacco users
Tobacco users, who reported using any form of tobacco daily or occasionally for more than 1 month prior to the interview.

### Sampling
#### Quantitative
All tobacco users identified in TNTS 2015/2016 in 11 purposively selected districts namely Chennai, Coimbatore, Kanchipuram, Madurai, Tirunelveli, Thiruvallur, Viluppuram, Pudukkottai, Kanyakumari, Tiruppur and Erode were recruited (n=2909) consecutively. These districts were purposively selected to ensure wider geographical coverage.

#### Qualitative
The sample for IDIs included a conveniently selected lot of tobacco users identified through TNTS 2015/2016, residing in Chennai, Kanchipuram and Thiruvallur Districts. A total of 26 IDIs were conducted. The participants of the telephone survey were divided into three groups: (1) those who made a failed quit attempt (n=10), (2) made a successful quit attempt (n=10) and (3) those who did not make any attempt (n=6). Around 6 to 10 IDIs were conducted in each of the three districts from three groups. Maximum variation sampling was used to include both smokers and smokeless tobacco users from different age groups. Data saturation was practiced using informational redundancy approach.[17] Further interviews

were discontinued if no new information was obtained pertaining to the major themes. However, there was inadequate response from the third group where the participants did not make any quit attempt.

### Analysis and statistics
#### Quantitative
Quantitative data were double entered and validated using EpiData entry (V.3.0) and analysed using EpiData analysis (V.2.2.2.183, EpiData Association, Odense, Denmark) and Stata V.13.0. The key outcome indicators were current tobacco use and quit attempt. $\chi^2$ test was used to find the association between various socio-demographic, tobacco use related variables with the current tobacco use. Binomial regression was used to explore the factors associated with tobacco use. Adjusted relative risks (aRRs) with 95% CIs was used to measure the strength of the association.

#### Qualitative
The audio-recorded interviews were transcribed manually in local language, Tamil, by the PI (SV) and the co-PI (RS) as soon as the interviews were over. The transcripts were read multiple times by two investigators (SV and RS) before coding. Thematic analysis following the six-phase approach by Braun and Clarke was undertaken to analyse the transcripts.[18] A hierarchical codebook was developed by two study investigators (SV and RS) by synthesising codes emerging directly from the transcripts (inductive) and from the topic guides (deductive). The initial coding was done independently by the investigators after going through the transcripts. The codes were then discussed and the discrepancies were resolved. Similar codes were combined to generate themes.[19] Verbatim are presented to support the findings. We have adhered to the Strengthening the Reporting of Observational studies in Epidemiology (STROBE) and Consolidated criteria for Reporting Qualitative research (COREQ) guidelines to report the study findings.[20]

Verbal informed consent was obtained from the participants by telephone. However, the calls were monitored by an individual not associated with the current research.

### Patients (participants) and public involvement
Participants were not involved in the design and conduct of the research, interpretation of results and writing of the manuscript. However, the study results will be disseminated to the participants and public by telephone calls/SMSs and newsletters. Simple short SMSs/messages will be developed in local language to disseminate the key findings of the study to the study participants. Newsletters in local language will be distributed to the patients and their relatives attending the Cancer Institute where the PI works.

### Data sharing statement
Extra data can be accessed via the Dryad data repository at http://datadryad.org/ with the doi: 10.5061/dryad.gtht76hj5.

**Table 1** Socio-demographic and characteristics of tobacco use among previously identified tobacco users in 11 selected districts during Tamil Nadu Tobacco Survey (2015/2016) who completed the follow-up survey in 2019 (n=555)

| Characteristics | N (%) |
| --- | --- |
| Current tobacco use | |
| Yes | 338 (60.9) |
| No | 217 (39.1) |
| Type of tobacco use | |
| Smoking | 243 (71.9) |
| Smokeless | 87 (25.7) |
| Both | 8 (2.4) |
| Type of tobacco smoke (n =) | |
| Cigarette | 151 (27.2) |
| Bidi | 121 (21.8) |
| Cigar | 01 (0.2) |
| Type of tobacco smokeless (n =) | |
| Tobacco chewing alone | 08 (1.4) |
| Tobacco + pan masala | 68 (12.3) |
| Snuff | 06 (4.5) |
| Others | 35 (6.3) |

## RESULTS

Of the 2909 tobacco users, only 724 (24.9%) could be contacted by telephone, of whom 555 (76.7%) consented for the interview. Of those consented, 210 (37.8%) were current tobacco non-users, while 337 (60.7%) were current tobacco users, remaining 8 (1.5%) had missing information. Of those who could not be contacted, the reasons for failing to contact were phone number not recorded (n=738, 33.8%), did not respond (n=109, 5.0%), expired (n=42, 1.9%) and other reasons (n=1296, 59.3%) such as number switched off, incorrect number, not reachable or not a valid number. (figure 1)

Socio-demographic and characteristics of tobacco use of the respondents are presented in table 1. Most of the respondents (511, 92%) were men. About 60.9% (n=338) were daily wage workers (who do not have a fixed occupation/salary but earn wages on a daily basis) followed by salaried individuals (government or private jobs, that is, those working in the private sector) and 44.3% (n=246) were educated up to secondary level. Majority of the respondents (243, 71.9%) were smokers. Table 2 compares the socio-demographic characteristics between those contacted versus those who could not be contacted by telephone. Significant difference in educational status was found between the groups (p=0.008).

As part of the qualitative component, a total of 26 IDIs were conducted. The socio-demographic details of the participants are given in table 3. Majority of them were men (22, 84.6%), belonging to the age group 45 to 59 years (12, 46.2%) and were daily wage labourers (15, 57.7%). The results of the thematic analysis were

**Table 2** Comparison of socio-demographic characteristics among those who could be contacted by telephone versus those who could not be contacted by telephone (n=2909)

| Characteristics | Contacted by telephone n=555 n (%) | Could not be contacted by telephone n=2354 n (%) | P value |
| --- | --- | --- | --- |
| Age | | | 0.1 |
| 18–24 | 11 (2.0) | 61 (2.6) | |
| 25–44 | 250 (45.0) | 1038 (44.1) | |
| 45–64 | 247 (44.5) | 1020 (43.3) | |
| ≥65 | 47 (8.5) | 235 (10.0) | |
| Gender | | | 0.06 |
| Male | 511 (91.8) | 2092 (90.6) | |
| Female | 44 (8.2) | 260 (9.4) | |
| Occupation | | | 0.12 |
| Unemployed: unable to work | 11 (2.0) | 71 (3.0) | |
| Unemployed: able to work | 12 (2.2) | 47 (2.0) | |
| Homemaker | 25 (4.5) | 151 (6.4) | |
| Daily wage | 338 (60.9) | 1349 (57.3) | |
| Self-employed | 82 (14.8) | 296 (12.6) | |
| Private/government job | 63 (11.4) | 299 (12.7) | |
| Missing | 24 (4.3) | 141 (6.0) | |
| Education | | | 0.008 |
| No formal school | 17 (3.1) | 106 (4.5) | |
| Primary | 105 (18.9) | 386 (16.4) | |
| Secondary | 246 (44.3) | 929 (39.5) | |
| Higher secondary and above | 86 (15.5) | 390 (16.6) | |
| Missing | 101 (18.2) | 543 (23.0) | |
| Intention to quit* | | | 0.1 |
| Yes | 338 (60.9) | 1522 (64.5) | |
| No | 148 (26.7) | 528 (22.6) | |
| Missing | 69 (12.4) | 304 (12.9) | |
| Exposure to smoke at home* | | | 0.09 |
| Yes | 362 (65.2) | 1452 (61.7) | |
| No | 185 (33.3) | 857 (36.4) | |
| Missing | 8 (1.5) | 45 (1.9) | |

*From previous Tamil Nadu Tobacco Survey.

categorised as: (1) barriers and (2) enablers of tobacco quitting which were further divided into three types: (i) intrinsic, (ii) extrinsic and (iii) support system. The themes emerged are presented as a thematic diagram (figures 2 and 3). The details of the themes, subthemes and verbatim quotes are presented in table 4.

**Table 3** Socio-demographic characteristics of participants of in-depth interviews, 2019

| Characteristics | Frequency | Percentage |
|---|---|---|
| Gender | | |
| Male | 22 | 84.6 |
| Female | 4 | 15.4 |
| Age | | |
| 18–24 | 2 | 7.7 |
| 25–44 | 9 | 34.6 |
| 45–59 | 12 | 46.2 |
| ≥60 | 3 | 11.5 |
| Occupation | | |
| Homemaker | 3 | 11.5 |
| Daily wage | 15 | 57.7 |
| Self-employed | 5 | 19.2 |
| Private/government job | 3 | 11.5 |
| Education | | |
| Primary | 2 | 7.7 |
| Secondary | 16 | 61.5 |
| Higher secondary and above | 5 | 19.2 |
| Missing | 3 | 11.5 |
| Quit attempt | | |
| Successful attempt | 10 | 38.4 |
| Failed attempt | 10 | 38.4 |
| Did not attempt | 6 | 23.1 |

Among those contacted and consented by telephone, 403 (72.6%) have made at least one attempt to quit, of whom 210 (52%) successfully quit and 193 (48%) made a failed quit attempt.

Among those who had quit successfully, we explored the enablers for quitting smoking which are described below.

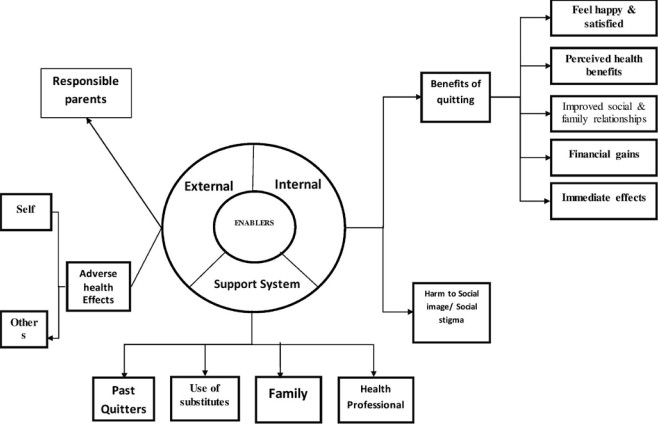

**Figure 2** Enablers of quitting tobacco and sustaining it among the tobacco users in three selected districts of Tamil Nadu, 2019.

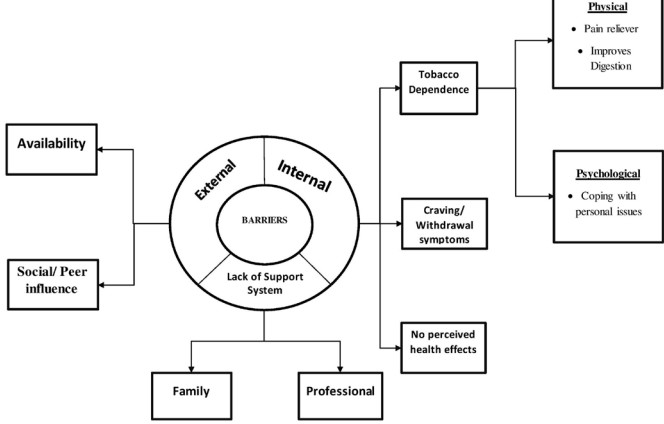

**Figure 3** Barriers of quitting tobacco and sustaining it among the tobacco users in three selected districts of Tamil Nadu, 2019.

### Enablers/motivators for quitting
#### Extrinsic factors
##### Adverse health effects
Recognition of the harms of tobacco to personal health and that of others, especially children in the family was reported to be a motivator for change.

> It is a bad habit, it causes many diseases, children do not like the habit. It is evident that our smoking habit affects others, it affects our health also. It can affect our health, cause cough, cold and cancer. (65, male)

##### Responsible parents
Some men spoke of their concerns about the harms from secondhand smoke and wanted to protect their children and family.

#### Intrinsic factors
##### Harm to social image
Upholding social image was considered as one of the key component of enablers which helped in successful quit attempt.

> When we smoke around women in a bus stop, they frown. They cover their face with a handkerchief while I smoke. I feel bad. How much ever we act decent, the respect for people who smoke is always less. People don't respect those who smoke. (56, male)

##### Benefits of quitting
Respondents found many advantages in quitting tobacco use such as being approved by their family, feeling contented from people's approval, financial benefits and improved health.

> An 65-year-old male opined, I was not able to eat much while using tobacco. Now I don't have any such feeling. Since I have quit, I am able to eat good amount of food. I don't have teeth stains and mouth ulcerations.

**Table 4** Themes and subthemes of enablers and barriers of quitting tobacco and sustaining it with corresponding quotes

| | Theme | Subtheme | | Quotes |
|---|---|---|---|---|
| Enablers | Extrinsic | Adverse health effects: Self and Others | | "It affects everything. It is a bad habit. It is harmful to health. I get cough, cold. All the internal organs are affected because of this. Quitting this is a very good deed" |
| | | Responsible parents | | "Doctors are saying that it affects the children immediately. All I want is children should not be affected, people at home should respect me and I should not have cough anymore. When my children said quit this, I decided to quit" |
| | Intrinsic | Harm to social image | | "People around us used to frown when we are using tobacco next to them. I used to think whether it is such a horrible thing" |
| | | Benefits of quitting | Immediate effects | "That is a very satisfying thing for me. I don't have any cough or cold after quitting" |
| | | | Feel happy and satisfied | "I am feeling good now. Because, I was addicted to a bad habit, but I have quit now. I feel that it's a good thing" |
| | | | Perceived health benefits | "Used to get cold, cough and would feel suffocated when smoking. Now after quitting, I am able to breathe normally. I am not getting exhausted now. I am able to feel that clearly. I am feeling happy that I quit" |
| | | | Improved social and family relationships | "I don't have cough. Now I can play with children. Initially I used to have a guilt that I keep coughing while playing with children" |
| | | | Financial gains | "When I am spending the 30 or 40 rupees from not purchasing cigarettes, for the sake of my children, I feel happy" |
| | Support system | Support from family | | "Family was very supportive. They always advise not to drink and not to smoke. Wife fights, daughter fights. It's a problem for everyone" |
| | | Support from past quitter | | "My friend advised that it would be beneficial to quit and that someone would be motivated to quit after seeing me" |
| | | Health advice by a doctor | | "Doctor advised me not to use this tobacco. I checked with him because I had burning sensation in the chest. Doctor said that it might be because of the tobacco that I use and advised me to reduce it" |
| | | Use of substitutes | | "I used to take tobacco after tea. Now as soon as I have tea I keep something in my mouth. I get the craving when I see people using tobacco, but I take vicks tablet at that time" |
| Barriers | Intrinsic | Tobacco Dependence | Coping with personal issues | "I smoke definitely when I am tensed. I smoke two to three cigarettes at a time when I am angry. If people make me angry, I will smoke to relax myself" |
| | | | Pain/stress reliever | "I use it occasionally, when I have toothache. Otherwise I won't. Only for toothache" |
| | | | Improves digestion | "I smoke only one cigarette after food. I use it for better digestion, that's it" |
| | | Casual usage: No perceived health effects | | "I will have such effects only if I use tobacco every day. But I use it only when I have toothache. So I don't have any effects" |
| | | Habitual user | | "What to do… Since it has become a habit for so long, I am unable to quit" |
| | | Craving: Withdrawal symptoms | | "When I am in the middle of a conversation, at times, I have this craving suddenly and I feel like I have to go immediately. I am unable to control the urge" |
| | Extrinsic | Availability of tobacco products | | "We are using this because it's available in the shops. Also, when someone smokes and exhales in front us, we get the craving" |
| | | Social/peer influence | | "Even if I stay at home trying to not use tobacco, I would want to use when someone who is using tobacco comes and says, just use it once" |
| | Support system | | | "If I am to quit, I will have to do it on my own will. Counselling or any sort of advice from others will not help in this case. Even when my family advices me, I move away from that place. I can quit, only if I make that decision on my own" |

Majority of the successful quitters reported that they had quit tobacco usage by their own will and determination (110, 50.7%), followed by advice from family (32, 14.7%) and advice from doctors (26, 12.0%). The least sought methods of cessation were counselling (3, 1.4%) and substitution (3, 1.4%) (table 5).

**Table 5** Method of cessation support sought (last attempt) to quit tobacco among those who are current non-smokers (n=210)

| Cessation method | N (%) |
| --- | --- |
| Counselling | 3 (1.4) |
| NRT | 5 (2.3) |
| Other medications | 16 (7.4) |
| Substitution | 3 (1.4) |
| Self (no support) | 183 (87.1) |
| Total | 210 (100) |

NRT, nicotine replacement therapy.

The qualitative interviews also revealed that support from family and advice by doctors were the enablers for quitting smoking besides the extrinsic and intrinsic personal motivators for quitting detailed above.

### Support system
#### Support from family
Support from children and spouse has been one of the positive reinforces which enabled successful quit attempts.

> A 65-year-old male said, My wife, son and friend were against this habit. So I decided to quit. Purely my decision and my wife's support.

#### Support from a past quitter
Support from successful quitters has helped the respondents quit tobacco use.

> My close friend had quit and he was supportive. He said it was good that I quit tobacco. (39, male)

#### Health advice by doctor
Advice by doctors also prompted many to quit the habit, especially those who already have adverse physical effects of tobacco.

> The doctor said if I continue smoking, I might die early. He advised me to quit and he said that all my internal parts have been affected to some extent. He also said that if I continue, I might get tuberculosis and other diseases. After that I felt that I should definitely quit. (56, male)

#### Use of substitutes
Respondents used substitutes such as chocolate, bubble gum and tulsi (*basil*) leaves to overcome craving and sustain the quit attempt.

Among current tobacco users, 115 (34.1%) did not make any quit attempt and 193 (57.3%) made a failed quit attempt. (figure 1)

Those who made a failed attempt or did not make any quit attempt were interviewed in-depth to explore the extrinsic, intrinsic and other barriers in quitting smoking which are narrated below.

### Barriers to quitting
#### Intrinsic factors
##### Tobacco dependence
Some current smokers talked about the ways in which smoking helped them 'cope' with adverse situations in life, such as giving comfort and relaxation at times of difficulties and thoughts to help manage personal tensions, work life problems and health issues. Consumption of tobacco allegedly helped the respondents to alleviate their pain or stress and improve digestion.

> Some or the other tension keeps happening. Some problem keeps occurring. At that time, when you smoke it is relaxing, feels good. Smoking one cigarette reduces anger. (46, male)

##### No perceived health effects
Some respondents were unaware of the consequences of long-term usage of tobacco while using it spontaneously without any specific intention.

> Health will be affected. We will become weak and have heavy breathing. But I do not do deep inhaling. I smoke very lightly and throw it away. So I think I don't have much effects. (67, male)

##### Craving: withdrawal symptoms
Respondents have reported strong urge to smoke and withdrawal symptoms such as nausea, vomiting, tingling sensation in mouth, headache and craving during the evenings after quitting.

#### Extrinsic factors
##### Availability of tobacco products
Many respondents opined that widespread availability of tobacco products makes it difficult to withhold them from usage.

> We are using because they are selling it. If they do not sell we won't use it. (36, male)

##### Social/peer influence
Some participants expressed that the offering of cigarettes from friends and relatives was the main reason for their failure to quit.

> Even if we stay at home wanting to stay away, when other people use, we get the craving. When others use and when they say smoke once nothing will happen, we get the urge. (46, male)

#### Support system
##### Lack of professional help
Some respondents have cited lack of professional help in terms of counselling or advice as a barrier to quit as they are not confident enough to do it on their own.

> A 45-year-old male respondent said, I am unable to do it on my own. I think counselling or any sort of

**Table 6** Association of socio-demographic and tobacco use related characteristics with current tobacco user status after the TNTS among previously identified tobacco users in 11 selected districts who completed the follow-up survey in 2019

| Characteristics | Total N | Current tobacco user n (%)† | Non-tobacco user n (%) | Unadjusted relative risk (95% CI) | P value | Adjusted relative risk (95% CI) |
|---|---|---|---|---|---|---|
| Age | | | | | | |
| 18–24 | 11 | 5 (45.5) | 6 (55.5) | 0.9 (0.5 to 1.9) | 0.8 | 0.8 (1.5 to 1.8) |
| 25–44 | 245 | 151 (61.6) | 94 (38.4) | 1.3 (0.9 to 1.7) | 0.13 | 1.3 (0.8 to 1.7) |
| 45–64 | 244 | 158 (64.8) | 86 (35.2) | 1.3 (1.0 to 1.8) | 0.05 | 1.3 (0.9 to 1.6) |
| ≥65 | 47 | 23 (48.9) | 24 (51.1) | 1.0 | – | 1.0 |
| Gender | | | | | | |
| Male | 509 | 313 (61.5) | 196 (38.5) | 1.0 (0.7 to 1.2) | 0.8 | – |
| Female | 38 | 24 (63.1) | 14 (36.8) | 1.0 | – | |
| Occupation | | | | | | – |
| Unemployed | 23 | 12 (52.2) | 11 (47.8) | 0.8 (0.5 to 1.3) | 0.4 | |
| Homemaker | 25 | 14 (56.0) | 11 (44.0) | 0.9 (0.6 to 1.4) | 0.6 | |
| Daily wage | 334 | 212 (63.5) | 122 (36.5) | 1.0 (0.8 to 1.3) | 0.8 | |
| Self-employed | 79 | 49 (62.0) | 30 (38.0) | 1.0 (0.8 to 1.3) | 0.9 | |
| Private/government job | 62 | 38 (61.3) | 24 (38.7) | 1.0 | – | |
| Previous tobacco use | | | | | | |
| Smoking | 395 | 251 (63.5) | 144 (36.5) | 1.2 (1.1 to 1.4) | 0.04 | 1.2 (1.1 to 1.4)† |
| Smokeless | 160 | 87 (54.4) | 73 (45.6) | 1.0 | – | |
| Previous intention to quit | | | | | | |
| Yes | 144 | 82 (56.9) | 62 (43.1) | 1.0 | – | 1.0 |
| No | 336 | 214 (63.3) | 122 (36.3) | 0.9 (0.8 to 1.1) | 0.09 | 0.9 (0.9 to 1.2) |
| Exposure to smoke at home* | | | | | | |
| Yes | 164 | 113 (68.9) | 51 (31.1) | 1.2 (1.1 to 1.4) | 0.008 | 1.2 (1.1 to 1.3)† |
| No | 313 | 182 (58.1) | 131 (41.9) | 1.0 | – | 1.0 |

*Captured during TNTS.
†Row percentage; education was removed because it had high multi-collinearity with occupation analysis has been adjusted for clustering at the district level.
TNTS, Tamil Nadu Tobacco Survey.

support would help. If possible, you can try to shut down the tobacco companies.

The quantitative survey also echoed this which said that counselling was the least sought method for cessation.

Significant association between current tobacco use and using smoking form of tobacco products (aRR=1.2, 95% CI: 1.1 to 1.4) and with exposure to smoke at home, which is a proxy indicator for smoking policy at home (aRR=1.2, 95% CI: 1.1 to 1.3) was noted (table 6).

## DISCUSSION

This is the first such study that we are aware of, to attempt a follow-up of participants of a survey done 3 years before by telephone calls to understand their current tobacco use status and whether they have made any quit attempt. Only one-fourth of the respondents could be contacted by telephone. This mixed-methods assessment among tobacco users of TNTS cohort found that of those

contacted and consented for telephone interview, one-third of them have successfully quit tobacco in the last 3 years and currently are non-tobacco users. Nearly three-quarters have made any quit attempt, of whom half of them could sustain the quit attempt. The qualitative part of the study identified the reasons for failure to quit and the enablers for quitting. The key findings of the study are discussed below.

Unsurprisingly, the study reported poor response rate to a telephone survey. Only one out of four respondents could be contacted. Although telephone surveys have been used widely in public health research and market research, there are concerns regarding poor response rate both due to failure to contact and refusal to participate once contacted. A major reason for poor response rate in this study could be the fact that the contact details of the study participants were collected nearly 3 years ago when the TNTS was conducted. It is highly likely that participants would have changed their numbers which is quite

common these days due to cut-throat competition in the telecom market and attractive offers by different network providers. Calls could not be made in a substantial proportion of cases, despite having a telephone number probably due to network issues, improper recording of phone number, tendency of people to switch between networks or possess more than one mobile number, and so on. Telephone number was not recorded in one-fourth of the respondents, meaning they either did not have any contact number/mobile phone or did not want to share the number or the number was not recorded. These considerations should be weighed in before planning any telephone survey. Moreover, different populations might have different challenges with respect to the use of telephone/mobile phone-based surveys, which needs to be understood before planning such surveys. Although telephone surveys yield poor response rates compared with household surveys which have response rates >90%, logistically telephone surveys are preferred.[21–23]

A study by Boland *et al* found poor response rate as low as 17.7% in telephone surveys similar to the present study.[24] In a community based telephone survey in the USA, response rate was 37%.[25] Another study in India in 2006 using telephone survey as a method of data collection yielded a high response rate of 94%. This was probably because it was a landline telephone-based survey and during those times landline numbers did not change frequently. The study was also done in a limited geographical area in urban location covering 50 households.[26] Based on the study experience and existing literature, we suggest additional strategies such as multi-modal data collection approaches instead of using single method, incentivisation and careful interviewer selection to improve response rate. In this study, the interviewer was a trained staff and part of a call centre of a project routinely involved in making telephone calls to project participants, native of Tamil Nadu (study area) and fluent in the local language (*Tamil*). However, nearly one-fourth of those who were contacted did not give consent for the interview, which requires additional intervention to improve participation. One such intervention was tried in Australia which concluded that mailing a postcard prior to the first telephone contact increases participation rate.[27]

One-third of the tobacco users have quit tobacco in the last 3 years and the remaining continue to use tobacco. This is an encouraging finding considering the poor quit rates of 5% to 10% across several studies.[7 8 28] However, this was self-reported and there was no objective way of assessing this response. A systematic review has shown trends of underestimation when smoking prevalence is based on self-report compared with cotinine-assessed smoking status.[29]

Nicotine addiction has been established the biggest cause of failure in smoking cessation. Tobacco dependence expressed in terms of craving for tobacco products, withdrawal symptoms, psychological dependence and habit forming emerged as the most important barriers to quitting in this study. These factors have specific management implications stressing the need for offering evidence-based tobacco cessation support including medications in line with the MPOWER strategy. The use of smoking cessation aids in our setting has been low similar to the findings of the present study. A national survey in India revealed that nearly 90% of former smokers quit without any professional aid.[30] Participants are reluctant to receive professional help and prefer to 'quit' by themselves. Few of the respondents also reported that quitting was difficult without support and were unaware of the availability of cessation aids. Evidence based tobacco cessation methods should be available and accessible to all through a primary care delivery model. People should be made aware of these services and their role in quitting tobacco and sustaining it.

Peer influence was a major barrier to quitting tobacco as reported in other studies as well.[31–33] Offering cigarettes/tobacco to one another is perceived as a sign of friendship and this culture serves as an impediment to smoking cessation. People need to be taught methods of rejecting the offer and that declining an offer of a cigarette/tobacco is not seen to be rude.

Most of the respondents reported symptoms of tobacco withdrawal during the initial phase of quitting. At the same time, unanticipated benefits such as a feeling of well-being both physically and psychologically, personal satisfaction, improved social relationships, encouragement from the family were also reported, and these benefits were 'self-reinforcing' in helping them to maintain their quit status. Thus, besides the health benefits, the collateral social, economic and psychological gains should also be conveyed to those who are interested in quitting tobacco as part of the counselling package.

The study found that tobacco users with a smoke-free policy at home were more likely to quit tobacco. This implies that smoke-free homes influence norms within the family around tobacco use. This inference could also be extended to other public places, thereby generating additional evidence for stricter implementation of smoke-free legislations in all public places.

The study investigators who conducted the IDIs are experienced qualitative researchers with strong interpersonal skills, which is essential in the context of telephone interview to establish rapport quickly and conduct interviews in a conversational manner. These skills helped the interviewer to work through tense and awkward moments that arose during the telephone interaction. Preparation of interviews was also done through mock trainings to handle any situation. The interviewers who work in a cancer care centre were not related to the participants nor were they involved in provision of their care directly or indirectly.

As far as we are aware, this is the first such attempt to reach out to tobacco users identified in the TNTS 2015/2016 after 3 years by a telephone survey. This novel method of survey gave useful insights into the use of telephone surveys in the Indian context and also provided

understanding related to quit attempts and successful quit rates in a large cohort of tobacco users.

The study had two key limitations. The major limitation was the poor response rate of the telephone survey opted due to resource limitation which might have introduced responder bias. However, the baseline characteristics of those who were contacted versus those who could not be contacted by telephone were similar except educational status, suggesting that the results could be generalised to the entire cohort. Second, there was no objective means of verifying the responses received by telephone survey. However, we feel that the social desirability bias is likely to be less in a telephone conversation due to lack of face-to-face interaction.

## Conclusion

Nearly two-thirds of the tobacco users have continued using it in the last 3 years. Lack of professional help and tobacco dependence were the major barriers to quitting which warrant decentralised evidence-based cessation interventions. There is evidence for the role of peer-led interventions involving family, peers and other tobacco users in quitting which could be incorporated into cessation interventions.

## Recommendations

Future research can consider on-field follow-up of tobacco users, as it could yield higher response rates than telephone follow-up. Research to increase response rates in a telephone survey can also be done. Considering the number of tobacco users who have quit or expressed their willingness to quit by their own self and determination, it is high time to develop interventions involving support system including family, friends and healthcare professionals as these were reported to be major catalysts facilitating quitting of tobacco.

**Author affiliations**
[1]Psycho-oncology, Cancer Institute-WIA, Chennai, Tamil Nadu, India
[2]Fenivi Research Solutions, Chennai, Tamil Nadu, India
[3]Community Medicine, All India Institute of Medical Sciences - Nagpur, Nagpur, Maharashtra, India
[4]Surgical Oncology, Cancer Institute-WIA, Chennai, Tamil Nadu, India
[5]Health Research, Indian Council of Medical Research, New Delhi, Delhi, India
[6]Community Medicine, Sri Devaraj Urs Medical College, Sri Devaraj Urs Academy of Higher Education and Research, Kolar, Karnataka, India
[7]Epidemiology, Bio-statistics and Cancer Registry, Cancer Institute (WIA), Chennai, Tamil Nadu, India
[8]Community Medicine, Yenepoya Medical College Hospital, Mangalore, Karnataka, India

**Acknowledgements** This research was conducted through the Structured Operational Research and Training Initiative (SORT IT), a global partnership led by the Special Programme for Research and Training in Tropical Diseases at the WHO (WHO/TDR). The model is based on a course developed jointly by the International Union Against Tuberculosis and Lung Disease (The Union) and Médecins Sans Frontières (MSF/Doctors Without Borders). The specific SORT IT programme which resulted in this publication was jointly developed and implemented by: The Union South-East Asia Office, New Delhi, India; the Centre for Operational Research, The Union, Paris, France; Médecins Sans Frontières (MSF/Doctors Without Borders), India; Department of Preventive and Social Medicine, Jawaharlal Institute of Postgraduate Medical Education and Research, Puducherry, India; Department of Community Medicine, All India Institute of Medical Sciences, Nagpur, India;

Department of Community Medicine, ESIC Medical College and PGIMSR, Bengaluru, India; Department of Community Medicine, Sri Manakula Vinayagar Medical College and Hospital, Puducherry, India; Karuna Trust, Bangalore, India; Public Health Foundation of India, Gurgaon, India; The INCLEN Trust International, New Delhi, India; Indian Council of Medical Research (ICMR), Department of Health Research, Ministry of Health and Family Welfare, New Delhi, India; Department of Community Medicine, Sri Devaraj Urs Medical College, Kolar, India; and Department of Community Medicine, Yenepoya Medical College, Mangalore, India. We would like to thank the recruiters and the telephone interviewers for the successful conduction of the study.

**Contributors** Conception and design and protocol development: SV, VGSE, JPT, NK, TA, RS. Data collection: SV, RS, DS. Data analysis: SV, JPT, SR, RS. Drafting the paper: SV, JPT, RS, TA, MM, AN. Critical review and final approval: SV, VGSE, JPT, AK, HRE, TA, NK, AN.

**Funding** The training programme, within which this paper was developed, was funded by the Department for International Development (DFID), UK.

**Competing interests** None declared.

**Patient consent for publication** Not required.

**Ethics approval** Ethics approval was obtained from the Institutional Ethics Committee, Cancer Institute (WIA), Chennai, Tamil Nadu, India, and the Ethics Advisory Group of the International Union Against Tuberculosis and Lung Disease, Paris, France.

**Provenance and peer review** Not commissioned; externally peer reviewed.

**Data availability statement** Data are available upon reasonable request. Extra data can be accessed via the Dryad data repository at http://datadryad.org/ with the doi: 10.5061/dryad.gtht76hj5

**ORCID iD**
Surendran Veeraiah http://orcid.org/0000-0002-6747-7423

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
