## [Reviewer comments · BMJ Open]

ARTICLE DETAILS

TITLE (PROVISIONAL)	Quit attempts amongst tobacco users identified in the Tamil Nadu Tobacco Survey of 2015-16: A 3 year follow-up mixed methods study
AUTHORS	Veeraiah, Surendran; Elangovan, Vidhubala; Tripathy, Jaya; Krishnamurthy, Arvind; Anand, T; M, Mahendra; Sudhakar, Revathy; K, Niraimathi; Subramani, Divyarajprabhakar; Rajaraman, Swaminathan; Elluswami, Hemanth Raj; Nirgude, Abhay

VERSION 1 – REVIEW

REVIEWER	Melanie Boeckmann Bielefeld University School of Public Health Germany
REVIEW RETURNED	16-Dec-2019

GENERAL COMMENTS	Dear colleagues Thank you for your research into quit attempts by adding a follow-up study component to a cross-sectional survey on tobacco use in India. I commend the use of a mixed methods design, and am commenting mainly on the qualitative study components. However, there are a number of points I think you need to address: Methods: Study design: Beyond the statement that you employed a sequential mixed methods design, no specific rationale for choice of this study design is given. I gather from your research question that the qualitative component was meant to explore in more depths why or why not quit attempts occurred, but this information is not reflected in the study design rationale. Please clarify why you chose this specific design, and what methodological/theoretical rationale guided you. Sampling: It is possible to have a convenience sample, yet it would help the readers if you explained why you chose these three districts for the interviews, and what was convenient about the 11 districts included in the telephone survey. This is important as there could be structural differences between districts shaping the responses you get.
--

	It is also not clear where you got the numbers of TNTS participants: had these been collected as part of the original survey? Please also give an indication of your planned sample size for the interviews. Were any of the respondents also included in the telephone survey. You mention data saturation: which approach to data saturation did you use, based on which literature? Data collection: I am missing information on why interviews were chosen as method to elicit the required information. Why not focus groups, or why not ask some open ended questions as part of the telephone survey? What added information were you looking for? This ties back into the question about choice of study design. Analysis: I am a bit surprised that you are citing not an original source for your approach to thematic analysis but an overview article (source 17) over types of thematic analyses. This means I cannot really tell whose approach to thematic analysis you planned to follow. Can you please clarify which approach you were using? Results: Could you please add a table giving the socio-demographic characteristics of your interview sample? Discussion: I would like to see some discussion about researcher reflexivity: what implications do you think the telephone survey design and the interviews had? How was the relationship between researchers and interview subjects in terms of power relations and similar aspects? What was the interview situation like? There is also no explicit data linkage/integration between the quantitative and qualitative components. Coming back to study design: what was the added value of conducting both analyses? How did the first quantitative component inform the interviews, and how and where did you integrate findings? Conclusion: This should (briefly) state how policy could address the barriers to cessation.
--	--

REVIEWER	Hannah Walsh King's College London, United Kingdom
REVIEW RETURNED	13-Mar-2020

GENERAL COMMENTS	Thank you for the opportunity to review this manuscript. Title: To increase impact, I suggest "Quit success (or "Quit attempts") amongst tobacco users identified in the Tamil Nadu Tobacco Survey of 2015-16: a 3 year follow-up mixed methods study" 2. The abstract is mainly concise and clear, but there are a few points which require amendment and/or clarification, as listed; Background para: Please state clearly that this study is a "follow-up" to the 2015-6 study; line 43 please explain or replace 'lakh' for
---

	an international readership; Results: when reporting the number of people it was possible to contact in the follow-up, please include all those who could be contacted, including those who declined to participate further, otherwise line 52 is misleading; 154, 55 needs clarification on number who actually quit vs number who made a quit attempt; 155 needs clarification as it is unclear whether using 'own will and determination' is a method of quitting, or a motivation to quit, same for 'advice from family'; 158 suggest replacing 'current' for 'continued' to stress the point these were 'non-quitters'. For remainder of this paragraph, clarify whether comments relate to those who quit or those who made quit attempt. In addition, comments on the INTRODUCTION: 195, 96: Please reference both of these statements; I am not sure there is good evidence to indicate that reducing (rather than quitting) is an effective way of saving lives, although it may have some impact on harm reduction; 1102: I think 'prevalence' instead of 'burden' is required; 1103 please indicate time frame for reference to "10% of all deaths...". The remainder of this paragraph would benefit from greater emphasis on the novelty of the follow-up to TNTS in order to justify the study. GATS is mentioned twice, further clarity required on its' limitations for this research question, as the phrase 'in different time periods' in 1109 is not sufficiently clear. Greater clarification and examples of the limitations of other similar studies (if they exist) would strengthen the case for this study; 1133 I suggest re-wording to "making and sustaining a quit attempt". 4. The methods section is comprehensive and would permit replication, but there are some points to address in order to aid clarity; Under 'general setting' it would be useful for the benefit of international readers to provide a description of what professional and/or pharmacological support for smoking cessation is available, i.e how widespread is this, how affordable is this, in order that the reader can interpret the results of quit methods used in context of what is available; 1150 please state how participants were invited i.e. by letter or phone call. Sample size: please include number of smokers identified in TNTS 2015-16, and use 'sampling frame' to differentiate from 'sample' 1196 please give rationale for how recording may affect participant response For qualitative methodology, please state whether any incentive was offered, and describe the context: if face to face, whereabouts? 1124: please give number instead of 'for some days' Please describe sampling method for qualitative interviews - were all smokers invited? Or was 30 the aim and recruitment continued until achieved ? 5. 1126 please state whether any issues were raised during the monitoring process PPI: please describe why PPI couldn't be carried out - this doesn't necessarily affect validity of study but for transparency please describe. Also, please clarify and expand plans to disseminate findings through phone calls and newsletters. 6. Please see earlier point re 1214. 8. Some statements in introduction and discussion require referencing, see separate points. 10. Tables 1+2 could be combined. Table 3 methods sought requires clarification - some of these could be considered motivations to quit rather than methods to quit such as pictorial warnings.
--	--

	I suggest fig 1a is not necessary. Fig 1b needs amending to demonstrate separately those who quit successfully and those who made a quit attempt. A description of participants in the qualitative interviews is required. 11. I397 is misleading - please give the proportion of those who could be contacted and who had reported quitting. The subsequent paragraph could include a statement as to whether similar studies have been carried out in Tamil Nadu. Consider whether these results are significantly different to other parts of the country, are there reasons why follow-up or low quit rate may be seen in Tamil Nadu compared to other regions of India? Greater emphasis could be placed on the relative novelty and uniqueness of this survey, which is one of its' great strengths. Comparisons with other follow-up surveys could be made which may indicate a relatively positive follow-up rate, considering factors pertinent and unique to this population. Boland et al is an Irish study - important to highlight that the two countries differ vastly, but nevertheless a similar rate was seen, it would be interesting to know your views on this. I415 It would be worth a brief discussion on why consent rate may have been low. Similarly, it would be useful to discuss the use of telephones - it may be a challenging method for reasons given, i.e. frequent change of number, lower than elsewhere mobile phone ownership? - all of which may indicate the follow-up rate was reasonable if not good given the challenges of this population. How does this compare with other (non-smoking)health related Indian health survey response rates? I427 please reference literature indicating validity of self-report vs biochemical verification of quitting I437 - 441 this appears to be referencing results which were not presented in the results section - please add these into results section, and discuss implications of results in discussion section I444 reference missing I466 Further exploration of educational status impact is warranted, as socio-economic status is a significant factor in smoking status in other countries and a comparative discussion is warranted. 12. Although brief reference is made to limitations in earlier bullet points, a paragraph in the discussion to aid interpretation of the findings is needed 15. Overall the standard is good, although there are some specific changes needed: use "by telephone", and "telephone survey". Please provide explanation of 'private job' and 'daily wage' for the benefit of international readers. Overall, this paper can make an important contribution to the literature, highlighting provision required for a population which is under-represented in research literature. There are a number of adjustments required to add clarity and therefore validity. The study would benefit from greater explanation of the context in order to demonstrate its' strengths.
--	---

VERSION 1 – AUTHOR RESPONSE

Reviewer: 1

Study design:

Beyond the statement that you employed a sequential mixed methods design, no specific rationale for choice of this study design is given. I gather from your research question that the qualitative component was meant to explore in more depths why or why not quit attempts occurred, but this information is not reflected in the study design rationale. Please clarify why you chose this specific design, and what methodological/theoretical rationale guided you.

Author's response: We have explained the rationale for adopting a sequential mixed methods design in the introduction section. (Line 139-146)

Sampling:

It is possible to have a convenience sample, yet it would help the readers if you explained why you chose these three districts for the interviews, and what was convenient about the 11 districts included in the telephone survey. This is important as there could be structural differences between districts shaping the responses you get.

Author's response: Thank you for the comment. The districts were selected purposively to cover all geographical corners of the state. We have revised the sampling strategy in the manuscript. (Lines 270-274)

It is also not clear where you got the numbers of TNTS participants: had these been collected as part of the original survey?

Author response: Yes, these were collected as part of the original survey. We have made it explicit in the study population and the sampling section. (Line 222)

Please also give an indication of your planned sample size for the interviews. Were any of the respondents also included in the telephone survey

Author's response: Yes the respondents were part of the telephone survey. The qualitative sample is a subset of the quantitative sample who were recruited by telephone survey. (Line 205)

You mention data saturation: which approach to data saturation did you use, based on which literature?

Author's response: Thank you for the comment. Data saturation was practiced using informational redundancy approach. We have made it clear in the sampling section. (Line 283)

Data collection:

I am missing information on why interviews were chosen as method to elicit the required information. Why not focus groups, or why not ask some open ended questions as part of the telephone survey? What added information were you looking for? This ties back into the question about choice of study design.

Author's response: These interviews were conducted telephonically, so focus groups were not feasible. The telephonic survey included mostly open ended questions to elicit responses from the respondents. (Line 250)

Analysis:

I am a bit surprised that you are citing not an original source for your approach to thematic analysis but an overview article (source 17) over types of thematic analyses. This means I cannot really tell whose approach to thematic analysis you planned to follow. Can you please clarify which approach you were using?

Author's response: We have cited an original source for thematic approach. (Line 301-303)

Results:

Could you please add a table giving the socio-demographic characteristics of your interview sample?

Author's response: We have added Table 5 describing the socio-demographic characteristics of the respondents who were interviewed as part of the qualitative component. (Line 729)

Discussion:

I would like to see some discussion about researcher reflexivity: what implications do you think the telephone survey design and the interviews had? How was the relationship between researchers and interview subjects in terms of power relations and similar aspects?

What was the interview situation like?

Author's response: Thank you for the comment. The in-depth interviews were conducted by telephone in a sub-set of the quantitative sample, who were vocal and willing to participate. We have discussed the relationship between the researcher and the interview subjects in the manuscript. (Line 557)

There is also no explicit data linkage/integration between the quantitative and qualitative components. Coming back to study design: what was the added value of conducting both analyses? How did the first quantitative component inform the interviews, and how and where did you integrate findings?

Author's response: The quantitative component estimates quit attempts and rates among tobacco users, whereas the qualitative component explores the barriers to quitting among those who did not make any quit attempt or made a failed quit attempt and enablers for quitting among those who made a successful quit attempt. The sample for the qualitative study is a sub-set of the quantitative sample.

Conclusion:

This should (briefly) state how policy could address the barriers to cessation.

Author's response: We have revised the conclusion to address the barriers to cessation. (Lines 577-580)

Reviewer: 2

Title: To increase impact, I suggest "Quit success (or "Quit attempts") amongst tobacco users identified in the Tamil Nadu Tobacco Survey of 2015-16: a 3 year follow-up mixed methods study"

Author's response: Thank you for the comment. Suggested change has been made in the title of the manuscript. (Line 3)

Abstract

Background para: Please state clearly that this study is a "follow-up" to the 2015-6 study;

Author's response: Suggested change has been made in the objective and methods section of the abstract. (Lines 45-52)

line 43 please explain or replace 'lakh' for an international readership;

Author's response: Suggested change has been made in the manuscript.

Results: when reporting the number of people it was possible to contact in the follow-up, please include all those who could be contacted, including those who declined to participate further

Author's response: Suggested change has been made in the results section of the manuscript. (Line 65-70)

line 52 is misleading; l54, 55 needs clarification on number who actually quit vs number who made a quit attempt.

Author's response: Suggested change has been made in the abstract for better clarity. (Line 67-70)

55 needs clarification as it is unclear whether using 'own will and determination' is a method of quitting, or a motivation to quit, same for 'advice from family';
Author's response: Thank you for the comment. We have made it clear that these are reported reasons for quitting tobacco, not any method of quitting.

58 suggest replacing 'current' for 'continued' to stress the point these were 'non-quitters'.
Author's response: Suggested change has been made in the results section. (Line 74)

For remainder of this paragraph, clarify whether comments relate to those who quit or those who made quit attempt.

Author's response: Suggested change has been made in the abstract. The adjusted analysis is exploring factors associated with continued tobacco use which includes those who made no attempt to quit or failed to make a successful quit attempt. (Line 74)

INTRODUCTION:

95, 96: Please reference both of these statements; I am not sure there is good evidence to indicate that reducing (rather than quitting) is an effective way of saving lives, although it may have some impact on harm reduction;

Author's response: Thank you for the comment. We have re-written the statement as "It is beyond any doubt that quitting tobacco is one of the most effective ways of saving lives and improving overall well-being". (Line 112)

102: I think 'prevalence' instead of 'burden' is required

Author's response: Suggested change has been made in the manuscript. (Line 118)

103 please indicate time frame for reference to "10% of all deaths...".

Author's response: Suggested change has been made in the manuscript. We have indicated the time frame as 2016. (Line 120)

The remainder of this paragraph would benefit from greater emphasis on the novelty of the follow-up to TNTS in order to justify the study. GATS is mentioned twice, further clarity required on its' limitations for this research question, as the phrase 'in different time periods' in 109 is not sufficiently clear. Greater clarification and examples of the limitations of other similar studies (if they exist) would strengthen the case for this study;

Author's response: Thank you for the comment. We have clarified the statement as suggested for better understanding of the readers. (Line 125-126)

133 I suggest re-wording to "making and sustaining a quit attempt".

Author's response: Suggested change has been made in the manuscript. (Line 156)

Methods

Under 'general setting' it would be useful for the benefit of international readers to provide a description of what professional and/or pharmacological support for smoking cessation is available, i.e. how widespread is this, how affordable is this, in order that the reader can interpret the results of quit methods used in context of what is available

Author's response: We have now described the tobacco cessation services available in the country for the benefit of the readers. (Lines 168-179)

150 please state how participants were invited i.e. by letter or phone call.

Author's response: The participants in the original TNTS survey were contacted by a door-to-door household survey. However, the participants for the present study were contacted by a telephone survey. This has been explained clearly in the manuscript. (Line 215)

Sample size: please include number of smokers identified in TNTS 2015-16, and use 'sampling frame' to differentiate from 'sample'

Author's response: Suggested change has been made in the sample size section. We have also changed the heading to sampling frame and sample size. (Line 202 & 205)

Please give rationale for how recording may affect participant response

Author's response: Thank you for the comment. We have stated the rationale for how recording may affect participant response as "The participants might be reluctant to share their experiences, if the calls are recorded, also referred to as Hawthorne effect." (Line 240)

For qualitative methodology, please state whether any incentive was offered, and describe the context: if face to face, whereabouts?

Author's response: Suggested change has been made in the methods section. (Line 256)

214: please give number instead of 'for some days'

Author's response: The definition used for current tobacco use is as follows "Tobacco users, who reported using any form of tobacco daily or occasionally for more than one month prior to the interview." Daily and occasional use are standard terms used in the Global Adult Tobacco Survey which is a globally accepted survey methodology for tobacco surveys. (Lines 266)

Please describe sampling method for qualitative interviews - were all smokers invited? Or was 30 the aim and recruitment continued until achieved?

Author's response: A total of 8-10 IDIs were planned to be conducted in each district to cover those who made a successful quit attempt, failed attempt and did not made a quit attempt. However, data saturation was practiced using informational redundancy approach to decide the sample size. Further interviews were discontinued if no new information was obtained pertaining to the major themes. This has been mentioned in the sampling section. (Lines 257-260 & 284-285)

216 please state whether any issues were raised during the monitoring process

Author's response: No major issues were raised, but feedback was given continuously to improve and fine-tune the process, especially during the initial phase of the study. (Line 243)

PPI: please describe why PPI couldn't be carried out - this doesn't necessarily affect validity of study but for transparency please describe. Also, please clarify and expand plans to disseminate findings through phone calls and newsletters.

Author's response: Suggested change has been made in the manuscript in the PPI section. We have also clarified the plans to disseminate findings through phone calls and newsletters. (Line 324)

Please see earlier point re 214.

Author's response: We have responded to the comment above.

Tables 1+2 could be combined.

Author's response: Thank you for the comment. Table 1 just describes tobacco use pattern of the respondents who could be contacted by telephone. However, Table 2 compares the characteristics of those who could be contacted by telephone versus those who could not be contacted by telephone. It does not compare the tobacco use pattern. This table has important implications in terms of response bias. So we feel that both the tables should not be combined as they convey different information. However, if the reviewer still feels that they could be combined, I suggest that we could remove Table 1 and write it as a narrative text in the results section.

Table 3 methods sought requires clarification - some of these could be considered motivations to quit rather than methods to quit such as pictorial warnings.

Author's response: Thank you for the comment. The methods of cessation depicted in the table are Counseling, NRT, Other medications, substitution and self. We have removed pictorial warning. (Lines 721)

I suggest fig 1a is not necessary.

Author's response: Thank you for the comment. We feel that this figure should be retained because it describes the SOP for making the telephone call and important for the readers to replicate. If you still feel it is not necessary, I suggest we can keep it as a supplementary appendix for the interested readers.

Fig 1b needs amending to demonstrate separately those who quit successfully and those who made a quit attempt.

Author's response: Suggested change has been made in Figure 1b.

A description of participants in the qualitative interviews is required.

Author's response: We have provided a separate Table 5 describing the characteristics of the participants who were interviewed. A brief description has also been provided in the results section. (Lines 367-369)

397 is misleading - please give the proportion of those who could be contacted and who had reported quitting.

Author's response: Thank you for the comment. We have revised the section as suggested. (Lines 461-463)

The subsequent paragraph could include a statement as to whether similar studies have been carried out in Tamil Nadu. Consider whether these results are significantly different to other parts of the country, are there reasons why follow-up or low quit rate may be seen in Tamil Nadu compared to other regions of India?

Author's response: This is the first such study in India to do a follow up assessment of survey participants. There is no precedence in the literature from the country, in fact a review of literature did not yield any similar study globally.

Greater emphasis could be placed on the relative novelty and uniqueness of this survey, which is one of its' great strengths. Comparisons with other follow-up surveys could be made which may indicate a relatively positive follow-up rate, considering factors pertinent and unique to this population. Boland et al is an Irish study - important to highlight that the two countries differ vastly, but nevertheless a similar rate was seen, it would be interesting to know your views on this.

Author's response: Thank you for the comment. We have mentioned it as a key strength of this study. We did not come across any similar follow-up surveys in a household survey setting using telephone. However, we have compared with other studies based on telephone survey. (Line 560-564)

The Boland et al. study was done more than a decade back in a completely different setting in Ireland. Moreover, the present study was done as part of 3-year follow-up of a survey which makes this comparison difficult. Despite this, the similar response rate in both the studies cannot be explained.

415 It would be worth a brief discussion on why consent rate may have been low.

Similarly, it would be useful to discuss the use of telephones - it may be a challenging method for reasons given, i.e. frequent change of number, lower than elsewhere mobile phone ownership? - all of which may indicate the follow-up rate was reasonable if not good given the challenges of this

population. How does this compare with other (non-smoking) health related Indian health survey response rates?

Author's response: We have tried to tease out the reasons for poor response rate in this telephone survey in the discussion section. (Lines 494-498)

427 please reference literature indicating validity of self-report vs biochemical verification of quitting

Author's response: We have added a systematic review indicating the underestimation of smoking status via self-report versus cotinine-assessed smoking status. (Line 517)

437 - 441 this appears to be referencing results which were not presented in the results section - please add these into results section, and discuss implications of results in discussion section

Author's response: Thank you for the comment. Peer influence was a key barrier to quitting tobacco as evident from the qualitative results already mentioned in the manuscript. (Line 400)

444 reference missing

Author's response: We have added reference as suggested. (Line 535)

Further exploration of educational status impact is warranted, as socio-economic status is a significant factor in smoking status in other countries and a comparative discussion is warranted.

Author's response: Thank you for the comment. The dependent variable which we have tried to model is current tobacco use among previously identified tobacco users i.e. unsuccessful quit attempt. Due to collinearity between educational status and occupation, only one of the variables had to be chosen for adjusted analysis. We chose occupational status due to some missing information in educational status. Therefore, the role of education could not be assessed. However, occupation was associated with education and therefore could be used as a proxy variable for education. Adjusted analysis showed that occupational status was not associated with current tobacco use status. Therefore, we have not discussed this relationship in this paper as this is not a key finding of this paper.

Although brief reference is made to limitations in earlier bullet points, a paragraph in the discussion to aid interpretation of the findings is needed

Author's response: Thank you for the comment. A paragraph on limitations is given in the discussion section. (Lines 566-573)

Overall the standard is good, although there are some specific changes needed: use "by telephone", and "telephone survey".

Author's response: Suggested corrections have been made throughout the manuscript.

Please provide explanation of 'private job' and 'daily wage' for the benefit of international readers.

Author's response: Thank you for the comment. We have clarified the terms 'private job' and 'daily wage worker' for the benefit of the readers. (Line 349-351)

VERSION 2 – REVIEW

REVIEWER	Melanie Boeckmann Bielefeld University School of Public Health, Germany
REVIEW RETURNED	28-Apr-2020
GENERAL COMMENTS	Dear colleagues thank you for addressing concerns and suggestions made in the previous review.

	There are a few remaining items to be addressed: Abstract: In the first sentence of Objectives, add a "to" before "explore the enables" Setting: missing % and n Participants: list how many interviews were conducted in the sub sample Methods: You write in your author's response that you gave rationale for sequential mixed methods design in lines 139-146, but I don't see it? It states why quantitative but not why sequential? Data saturation: please cite a reference for the informational redundancy approach Results: As in my previous review I still believe that for a mixed methods design (rather than multimethod) there needs to be integration of data or findings. Your rationale for this could be added to the strength section.
--	--

REVIEWER	Hannah Walsh King's College London, UK
REVIEW RETURNED	05-May-2020

GENERAL COMMENTS	Thank you for the opportunity to review this important manuscript. I note the changes made, which have addressed the suggested changes, and the enhanced discussion in this draft. There are a few minor points I would like to comment on, see below. Overall the paper would benefit from a review and edit of English usage, there are some adjustments to be made for clarity and some for style purposes which would enhance the paper. Fig 1a: I suggest this is included in supplementary materials 1st strength: I suggest including '...that we are aware of...' as a note of caution I154 "behavioural support" I218 I wouldn't consider the explanation necessary, for brevity include the monitoring purpose I433 As noted earlier, suggest 'As far as we are aware...' I460 The figure needs referencing
--

VERSION 2 – AUTHOR RESPONSE

Reviewer 1

Dear colleagues

Thank you for addressing concerns and suggestions made in the previous review. There are a few remaining items to be addressed:

Author's response: Thank you for your valuable comments. We have tried to address your concerns in a point-wise response below.

Abstract:

In the first sentence of Objectives, add a "to" before "explore the enables"

Author's response: Suggested change has been made in the manuscript. (Line 40)

Setting: missing % and n

Author's response: Suggested change has been made in the manuscript. (Line 45)

Participants: list how many interviews were conducted in the sub sample

Author's response: A total of 26 in-depth interviews as part of the qualitative component. We have made it explicit in the abstract and in the qualitative sampling section. (Line 49 & 266)

Methods: You write in your author's response that you gave rationale for sequential mixed methods design in lines 139-146, but I don't see it? It states why quantitative but not why sequential?

Author's response: Thank you for the comment. We have now stated why a sequential approach was adopted. (Line 150)

Data saturation: please cite a reference for the informational redundancy approach

Author's response: Suggested change has been made in the manuscript. We have added a suitable reference. (Line 272)

Results:

As in my previous review I still believe that for a mixed methods design (rather than multimethod) there needs to be integration of data or findings. Your rationale for this could be added to the strength section.

Author's response: Thank you for your remark. We have now integrated the findings of the quantitative and qualitative components so that they complement each other. (Line 339- 450)

Reviewer: 2

Please leave your comments for the authors below

Thank you for the opportunity to review this important manuscript. I note the changes made, which have addressed the suggested changes, and the enhanced discussion in this draft.

Author's response: Thank you for your encouraging remarks. We have tried to address your remaining concerns in a point-wise response below.

There are a few minor points I would like to comment on, see below.

Overall the paper would benefit from a review and edit of English usage, there are some adjustments to be made for clarity and some for style purposes which would enhance the paper.

Fig 1a: I suggest this is included in supplementary materials

Author's response: Thank you for the comment. We have included Figure 1a as supplementary material under the heading Figure 1. (Line 216)

1st strength: I suggest including '...that we are aware of...' as a note of caution

Author's response: Thank you for the comment. We have reworded the statement as "This is the first such study that we are aware of, to attempt a follow up....." (Line 74)

l154 "behavioural support"

Author's response: Suggested change has been made in the manuscript. (Line 163)

l218 I wouldn't consider the explanation necessary, for brevity include the monitoring purpose

Author's response: Suggested change has been made in the manuscript. We have removed the explanation, but retained the monitoring part. (Line 229)

l433 As noted earlier, suggest 'As far as we are aware...'

Author's response: Thank you for the comment. We have reworded the statement as "As far as we are aware, this is the first such attempt to....." (Line 453)

l460 The figure needs referencing

Author's response: Thank you. The references for the figure is added to the manuscript.

VERSION 3 – REVIEW

REVIEWER	Melanie Boeckmann Bielefeld University School of Public Health
REVIEW RETURNED	29-May-2020
GENERAL COMMENTS	Thank you for addressing my concerns